# Increased Risk of Dentoalveolar Trauma in Patients with Autism Spectrum Disorder: A Systematic Review with Meta-Analysis

**DOI:** 10.3390/ijerph21121563

**Published:** 2024-11-26

**Authors:** Izabela da Costa, Rafael Binato Junqueira, Daniele Sorgatto Faé, Luisa Amorim Pêgas de Souza, Cleidiel Aparecido Araujo Lemos

**Affiliations:** 1Department of Dentistry, Federal University of Juiz de Fora (UFJF), Campus Avançado Governador Valadares, Governador Valadares 35010-180, MG, Brazil; izabela.costa@estudante.ufjf.br (I.d.C.); rafael.binato@ufjf.br (R.B.J.); luisa.amorim@estudante.ufjf.br (L.A.P.d.S.); 2Postgraduate Program in Applied Health Sciences (PPgCAS), Federal University of Juiz de Fora, Governador Valadares 35010-180, MG, Brazil; daniele.fae@ufjf.br

**Keywords:** autistic disorder, special need, dental trauma, dentoalveolar trauma

## Abstract

The prevalence of traumatic dental injuries (TDI) in patients with autism spectrum disorders (ASD) remains unclear. Given these discrepancies, an updated review of the evidence on the risk of TDI in patients with ASD is essential. This systematic review and meta-analysis aimed to evaluate the prevalence of TDI in patients with ASD and compare it to that in neurotypical patients. This study protocol was registered on PROSPERO (CRD42024580127) and followed the Cochrane Handbook for Systematic Reviews of Interventions and PRISMA guidelines. A comprehensive search of four databases—MEDLINE/PubMed, Web of Science, Scopus, and Embase—was conducted for articles published up to August 2024. Moreover, the gray literature (ProQuest) and reference lists were screened. The inclusion criteria required participants with ASD to assess TDI across deciduous, mixed, and permanent dentition regardless of age. No restrictions were applied on TDI type, language, or publication date. Additionally, case reports, reviews, letters, and studies addressing other oral disorders without specific TDI data were excluded. A single-arm meta-analysis evaluated the cumulative proportion and 95% confidence interval (CI) of TDI in patients with ASD. Moreover, a comparative meta-analysis was performed to assess the risk of TDI between ASD and neurotypical patients, calculating the odds ratio (OR) with a 95% CI, and a *p* < 0.05 was deemed significant, using the R program. Quality assessment was performed using the National Heart, Lung, and Blood Institute tool, and the certainty of evidence was evaluated using GRADE. A total of 22 studies were included to determine the overall prevalence of TDI, of which 16 studies directly compared patients with ASD to neurotypical individuals. In total, 3817 participants were evaluated, including 2162 individuals with ASD and 1655 neurotypical patients. A single-arm meta-analysis estimated a TDI prevalence of 22% (Confidence Interval [CI]: 17–27%) among patients with ASD. A significant difference in the risk was observed between ASD and neurotypical patients (*p* = 0.003; Odds Ratio [OR]: 1.67; CI: 1.19–2.26). However, substantial heterogeneity was observed in this analysis. Although the majority of studies were rated as high quality, the certainty of the evidence was considered very low. Despite the limitations of this study, the findings suggest that patients with ASD are at a higher risk of developing TDI than the risk observed in neurotypical patients. Therefore, preventive educational initiatives are recommended to reduce the risk of TDI in this population.

## 1. Introduction

Autism spectrum disorder (ASD) is a neurodevelopmental condition characterized by deficits in verbal and nonverbal communication, impaired social interactions, and repetitive or stereotypical behaviors. The condition typically manifests during early childhood and persists until adolescence and adulthood [1]. Although the exact etiology of ASD remains unknown, the prevalence estimates vary. Recent studies have indicated that approximately 1 in 100 individuals is affected by ASD [2]. In the United States of America, where diagnostic methods and data mapping are advanced, the prevalence may be as high as 1 in 32 [3], reflecting a steady increase in diagnoses in recent years [4].

Patients with ASD often experience dental issues that negatively affect their oral health and overall quality of life [5]. Studies have demonstrated that individuals with ASD have limited access to dental care, which further compromises their oral health [6,7]. Many individuals encounter significant barriers to accessing oral health services, with a high percentage of children with ASD having never visited a dentist [8].

Among the various oral manifestations that affect patients with ASD, dental trauma is a significant concern. Dental trauma is a significant public health issue caused by the transmission of external forces to the dental tissues or adjacent structures, particularly in children and adolescents [9]. Many traumatic dental injuries (TDI) could be preventable by following specific recommendations, such as avoiding parafunctional habits (e.g., bruxism or chewing on foreign objects like ice, paper clips, or pens) and refraining from using intraoral ornaments. Additionally, the use of mouthguards during contact sports and addressing dental misalignment, particularly of protruding upper teeth, are effective strategies for decreasing the incidence of TDI [10].

Nevertheless, individuals with special needs, including those with ASD, are at an increased risk of dental trauma due to cognitive, psychomotor, and behavioral impairments [11]. Impulsivity, self-harm tendencies, anger, and tantrums, which patients with ASD often exhibit, further increase this risk [11,12]. In a systematic review, Silveira et al. [5] evaluated the relationship between patients with special needs and TDI and discovered that these patients generally exhibited an elevated risk of TDI. However, the authors reported no significant difference in TDI risk between patients with ASD and those in the healthy group. This may be attributed to the limited number of studies (only six) that included patients with ASD. Conversely, a substantial body of the literature suggests that individuals with autism have an elevated risk of experiencing TDI [13,14,15]. However, some studies have reported a higher risk among neurotypical patients compared to those with ASD [9,16]. Given these conflicting findings, an updated review of the available evidence regarding the risk of TDI in patients with ASD is warranted. Therefore, this systematic review and meta-analysis aimed to evaluate the prevalence of TDI in patients with ASD and determine whether this group is at a higher risk than the risk observed in neurotypical patients. The hypotheses tested were (1) that ASD does not influence the prevalence of TDI and (2) that the incidence of TDI in patients with ASD would be comparable to that in neurotypical individuals.

## 2. Materials and Methods

### 2.1. Protocol and Registration

This review was conducted according to the Cochrane Handbook for Systematic Reviews of Interventions [17] and adhered to the Preferred Reporting Items for Systematic Reviews and Meta-Analyses (PRISMA) guidelines [18]. A detailed methodological protocol outlining the necessary steps to be undertaken was developed and registered in the International Registry of Systematic Reviews (PROSPERO–CRD42024580127). The PRISMA-P extension [19] was employed to guide the protocol development.

### 2.2. Eligibility Criteria

A focused and structured research question for this systematic review was developed based on the PECOS framework (Population; Exposure; Comparison; Outcomes; and Study Design). The question addressed was, “Are individuals with ASD at increased risk of TDI compared to those without ASD?”
Population: Patients of all ages—children, adolescents, and adults—who had experienced dental injuries resulting from TDI.Exposition: Patients with ASD.Comparison: Neurotypical patients.Outcome: Prevalence of TDI in patients with ASD and neurotypical patients.Study Design: Observational studies (cohort, cross-sectional, or case-control studies).

The inclusion criteria specified that participants must have a definite diagnosis of ASD, and assessments of TDI encompassed deciduous, mixed, and permanent dentition regardless of age. No restrictions were imposed regarding the type of TDI (e.g., enamel fractures, dentin fractures, root fractures, alveolar process fractures, subluxations, lateral dislocations, intrusive or extrusive dislocations, and avulsions). Moreover, no limitations were placed on language or publication date. Case reports, reviews, letters to the editor, and studies focusing on other oral disorders without proper reporting of TDI incidence were excluded.

### 2.3. Search Strategies

The literature search was carried out independently by two authors (I.C. and D.S.F.), who were calibrated on the search process and application of eligibility criteria to identify potentially eligible studies. Disagreements were resolved by a third reviewer (C.A.A.L.). Searches were performed using various electronic databases, including MEDLINE via PubMed, Web of Science, Embase, and Cochrane Library. The search strategies were subjected to a peer-review process to ensure adherence to high-quality standards for study selection [17], using the “Peer Review of Electronic Search Strategies” (PRESS) tool [20]. The detailed search strategies for each database are provided in Appendix A. A Rayyan QCRI reference manager was utilized [21]. The workflow involved importing search results, removing duplicates, selecting relevant studies, and resolving disagreements. In addition to electronic database searches, the gray literature was explored using ProQuest and ClinicalTrials.gov. A manual search of the reference lists of eligible articles was also performed.

### 2.4. Data Extraction

Data from the identified studies were collected by one reviewer (I.C.) and verified by a second reviewer (C.A.A.L.) to minimize potential errors during the extraction process. Additionally, data were extracted using a preliminary extraction form developed as part of the methodological protocol for this systematic review. When essential information for extraction was unavailable, the corresponding authors of the studies were contacted to obtain the necessary details.

Common Office suite tools, including Microsoft Word, were employed for data extraction. Information collected from each study was systematically analyzed to ensure data uniformity. The initial parameters established for data collection from the studies included the following: (1) Author/Year; (2) study design; (3) country; (4) number of patients (autism and neurotypical); (5) gender; (6) age (mean or range); (7) setting of study; (8) trauma classification; (9) prevalence of TDI (%) calculated based on the reported incidence of TDI and the number of patients with ASD or neurotypical; (10) key findings as reported in the results and conclusions of the included studies.

### 2.5. Quality Assessment 

The quality of the selected studies was assessed using the quality assessment tool provided by the National Heart, Lung, and Blood Institute. This tool consists of nine questions that ultimately categorize this study’s quality as “good”, “fair”, or “poor”, with “good” indicating high quality. Furthermore, studies scoring at least 7 points are deemed satisfactory. One reviewer (I.C.) performed the quality assessment, and a second reviewer (C.A.A.L.) verified the tabulated findings to ensure consistency.

### 2.6. Data Synthesis

A single-arm meta-analysis evaluated the cumulative proportion and 95% confidence interval (CI) of TDI in patients with ASD. Additionally, a comparative meta-analysis was performed to assess the risk of TDI in both patients with ASD and neurotypical individuals. The analysis focused on the odds ratio (OR) with a 95% CI, and a *p*-value < 0.05 was deemed statistically significant. A random-effects model was applied, and a sensitivity analysis was performed in cases of significant heterogeneity. The R software (version 4.4.2 for Windows) was used for the meta-analysis, utilizing the “meta” and “metafor” packages.

### 2.7. Certainty of Evidence and Additional Analysis

The certainty of evidence was evaluated using the Grading of Recommendations Assessment, Development, and Evaluation (GRADE) framework. This approach enables the assessment of evidence reliability for each outcome by considering factors such as study design, inconsistency, indirectness, imprecision, and the risk of publication bias. Finally, the certainty of each outcome was rated as high, moderate, low, or very low. The Summary of Findings tables were generated using GRADEpro GDT software (https://gdt.gradepro.org/app/) [22]. Additionally, an analysis was performed to assess the agreement among examiners during the individual study selection process, utilizing the kappa concordance criteria [23].

## 3. Results

### 3.1. Study Selection

The search process yielded 167 articles, 65 of which were sourced from PubMed/MEDLINE, 63 from Embase, 20 from the Web of Science, and 4 from the Cochrane Library. Additionally, a search for the gray literature was conducted using ProQuest, ClinicalTrials.gov, manual searches on websites, and a reference list of the included studies, totaling 15 articles. After the elimination of 70 duplicate entries, 97 studies were selected for further analysis. Of these, 23 were deemed eligible for full-text reading. One study was excluded as the research focused exclusively on patients with traumatic conditions in both patients with ASD and neurotypical patients [24]. Thus, 22 articles were included in the analysis [9,14,15,16,25,26,27,28,29,30,31,32,33,34,35,36,37,38,39,40,41,42]. A high level of agreement was achieved in the selection of articles, with a kappa value of 0.83, indicating substantial consistency based on the kappa criteria [23]. The complete search strategy is visualized as a flow diagram (Figure 1).

### 3.2. Characteristics of Included Studies

A total of 22 studies were included in the review [9,14,15,16,25,26,27,28,29,30,31,32,33,34,35,36,37,38,39,40,41,42]. Of these, 16 studies included a direct comparison between individuals with ASD and neurotypical patients regarding the occurrence of TDI [9,14,15,16,25,26,27,28,29,31,34,35,38,39,40,42], whereas the remaining 6 studies [30,32,33,36,37,41] focused solely on the prevalence of TDI among ASD without the inclusion of the neurotypical patients. These studies were published between 2009 and 2024, predominantly employed with the most common study design being cross-sectional study design. These studies spanned a diverse range of countries, with Brazil and India being the most represented. In total, 3817 participants were assessed, including 2162 individuals with ASD and 1655 neurotypical patients. Participants ranged from children and adolescents (aged 3–18 years) to adults (aged 20–41 years), with the majority of these studies focusing on school-aged children, encompassing the primary, mixed, and permanent dentition stages.

These studies were performed across various settings, most commonly in dental schools, clinics, and hospitals. In terms of dental trauma classification, two studies applied the Ellis fracture classification [30,33], whereas four studies used the World Health Organization (WHO) system modified by Andreasen [29,31,34,42]. The remaining studies did not specify the classification method used.

Some studies have detailed the types of TDI encountered, including fractures of enamel, dentin, and roots, pulp involvement, and various forms of luxation (concussion, subluxation, intrusive, and lateral) and avulsions. Common causes of trauma include falls [14,31,33], traffic accidents [33], self-injury [14,31], and other unknown causes [14] (Table 1).

### 3.3. Meta-Analysis

All included studies were pooled in a single-arm meta-analysis to estimate the prevalence of TDI among individuals with ASD. Using a random-effects model, the overall prevalence of trauma in these participants was estimated to be 22% (CI: 17% to 27%). The analysis revealed considerable heterogeneity among the studies (I^2^ = 87%, *p* < 0.01) (Figure 2).

In studies comparing TDI between patients with ASD and neurotypical, a statistically significant difference was identified, with patients with ASD demonstrating a high risk of TDI (*p* = 0.003; OR: 1.67; CI: 1.19 to 2.26). This analysis indicated substantial heterogeneity (I^2^ = 63%, *p* = 0.0003) (Figure 3). The source of this heterogeneity appeared to stem from two specific studies [9,16] that showed a high incidence of TDI among neurotypical patients compared to those with ASD patients. After conducting a sensitivity analysis and removing these two studies, the findings still indicated a significantly higher incidence of TDI in ASD participants (*p* < 0.0001; OR: 1.87; CI: 1.46 to 2.38). However, the heterogeneity decreased to a non-significant level (I^2^ = 31%, *p* = 0.13) (Figure 4).

### 3.4. Quality Assessment and Certainty of Evidence

In terms of quality assessment, most of the included studies were of high quality, with scores of seven or higher. Specifically, four studies scored 9 points, two studies scored 8 points, seven studies scored 7 points, two studies scored 6 points, and five studies scored 5 points. Table 2 provides a detailed explanation for the low scores in certain studies. The certainty of evidence evaluated using the GRADE approach was very low for the comparison of TDI between patients with ASD and neurotypical patients. The initial certainty was rated low, given that only observational studies were included, and it was further downgraded owing to indirectness. No upgrades were applied for large effects, plausible confounding factors, or dose-response gradients.

## 4. Discussion

This study aimed to assess the incidence of TDI in patients with ASD and compare it to the incidence in neurotypical patients. The first and second hypotheses were rejected, as direct comparative analysis revealed a significantly high probability of TDI in patients with ASD. According to a previous study, the global prevalence of TDI is approximately 10–15% [43]. However, our systematic review indicated a TDI rate of 22% (CI: 17–27%) among patients with ASD. These findings suggest that patients with ASD are at a significant risk of TDI, especially when compared with the risk in neurotypical patients. Notably, only one study [9] reported a lower incidence of TDI in patients with ASD compared to the incidence observed in neurotypical patients. The authors attributed this to ASD-related difficulties in social interaction and impaired communication, which may lead to patient isolation and reduced exposure to common causes of TDI [9]. However, other studies have demonstrated that TDI can occur frequently in patients with ASD, even during routine daily activities, such as falls while walking [13,26,29]. Increased incidence of falls during routine activities in individuals with ASD may be linked to delayed cognitive development and impaired motor coordination [13,14,34]. This aligns with the findings of Habibe et al. [14], who reported high TDI rates in patients with ASD during routine activities, whereas in neurotypical patients, TDI occurred mainly during collective or leisure activities.

Additionally, factors including muscular incoordination, altered muscle tone [24], and common oral habits such as thumb sucking and object biting contribute to altered occlusion and facial patterns in children with ASD [34]. Frequent malocclusions in patients with ASD include open bite [16,24,26], poor lip closure, maxillary incisor protrusion [24,26], and increased overjet [13]. These factors, individually or in combination, may further increase susceptibility to TDI.

Another significant factor is self-injurious behavior [13,29,33,44]. Many patients with ASD exhibit episodes of agitation, hyperactivity, anxiety, obsessive–compulsive tendencies, and psychotic/personality disorders, all of which contribute to aggressive or self-injurious behaviors [45]. These behaviors often target the orofacial regions and can range in intensity from mild to severe head banging, potentially explaining the high TDI rates in this group [34].

Despite these findings, this study had several limitations that must be acknowledged. None of the included studies provided detailed diagnostic criteria for ASD. The use of various diagnostic systems, including the Diagnostic and Statistical Manual of Mental Disorders (DSM-5) and the International Classification of Diseases (ICD-11), introduces global variability in diagnosis. For instance, the ICD-11 combines previously distinct conditions, such as Asperger syndrome, into a broad ASD category. Regional differences in the interpretation and application of these guidelines, especially in resource-limited settings or different cultural contexts, can lead to disparities in the diagnosis and management of ASD [46]. This highlights the need for a universally standardized diagnostic protocol to evaluate ASD.

Another limitation is that only a few studies [14,26,29,31,32,34,42] specified the type of sustained TDI, while only three [14,33,34] identified the affected teeth. Such detailed results are crucial for understanding the TDI dynamics. For instance, fractures of enamel and dentin typically involve the upper incisors owing to their position in the dental arch [24,34]. Thus, implementing preventive strategies tailored to patients at high risk of TDI is essential to mitigate the occurrence of such injuries.

An age- or gender-based analysis of the TDI in patients with ASD was not feasible due to limited data. Most of the included studies reported only the age range of participants without detailing TDI incidence within specific age groups. Regarding gender, some studies identified no significant differences between the genders [28,31,36,37,42], while others reported higher rates of TDI in males [29,33] or females [9,14]. This inconsistency may result from the high proportion of male patients with ASD in the included studies, which is consistent with the literature indicating a 2–3 times higher prevalence of ASD in males [45]. Future studies should consider variables such as type of TDI, affected teeth, age, and sex to elucidate these differences.

Knowledge gaps regarding ASD contribute to underdiagnosis and complicate the analysis of population studies. As previously highlighted, the study by Andrade et al. [9] in Brazil was the only study to report significantly more TDI in neurotypical patients than patients with ASD. Notably, 70% of the participants in this study were from low-income backgrounds, which may have affected their access to healthcare and diagnosis [14,47]. Furthermore, 71.4% of families with children with ASD did not seek dental care following trauma, which may have contributed to the underreporting of TDI in patients with ASD. A recent cross-sectional study conducted in Brazil discovered that 25% of children with ASD had never visited a dentist [48]. This high prevalence of children with ASD who have never received dental care has also been documented in studies from other countries [49,50,51]. This finding is corroborated by Andreasen et al. [52], who reported that the prevalence of TDI is influenced by socioeconomic, behavioral, and cultural factors. This underscores the regional influence of the aforementioned factors and the necessity for experienced professionals to promptly address TDI, particularly in vulnerable populations, such as patients with ASD [53].

Considering the occurrence of TDI in clinical practice, investigating its incidence in patients with ASD is essential. These insights could support the development of preventive and intervention strategies to reduce the frequency and severity of TDI [33]. Finally, the high heterogeneity of the included studies warrants a cautious interpretation of the findings. Differences in study design, geographic region, population selection, eligibility criteria, diagnostic methods, inherent biases in retrospective studies, and the varying quality of the included studies contributed to this variability. Therefore, identifying patients at a high risk of developing TDI by observing behavioral risk factors is crucial. Future well-conducted studies are recommended to reevaluate the findings of this systematic review.

## 5. Conclusions

Considering these limitations, this study confirmed a TDI prevalence of 22% among patients with ASD, with a significantly higher incidence of TDI compared to neurotypical patients. These findings underscore the urgent need for targeted preventive strategies and educational initiatives to reduce the TDI risk in vulnerable populations.

## Figures and Tables

**Figure 1 ijerph-21-01563-f001:**
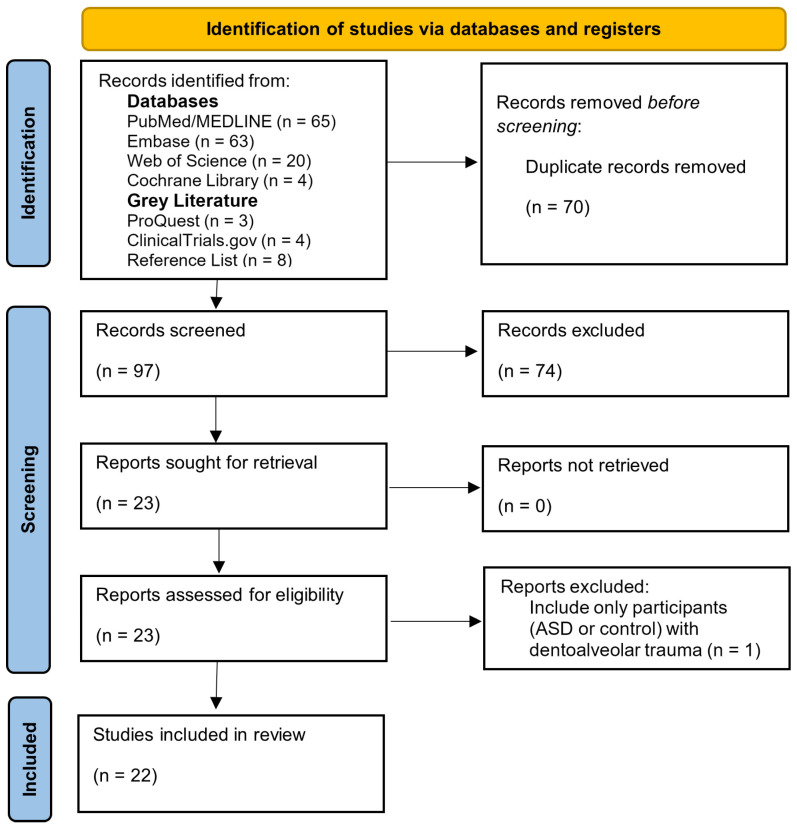
Flow diagram illustrating the search strategy and selection process.

**Figure 2 ijerph-21-01563-f002:**
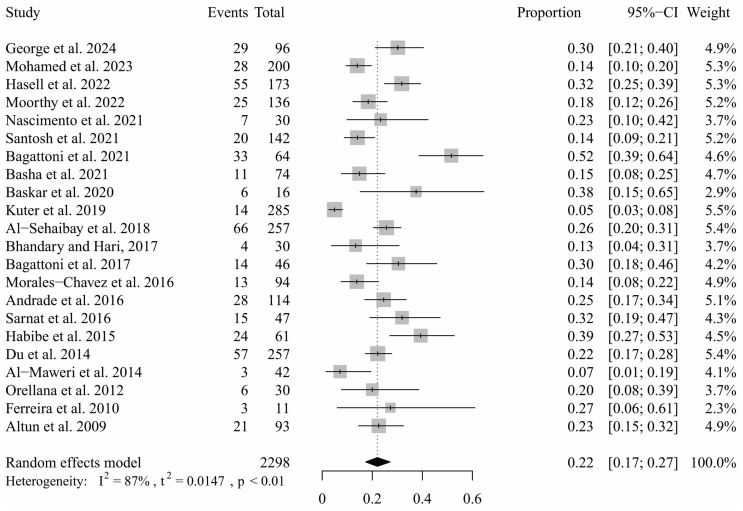
Single-arm meta-analysis drawing the proportion of TDI in patients with ASD.

**Figure 3 ijerph-21-01563-f003:**
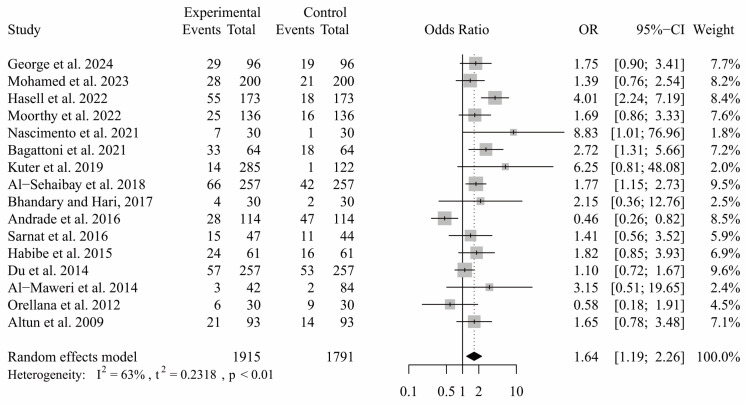
Meta-analysis comparing the incidence of TDI between patients with ASD and neurotypical patients.

**Figure 4 ijerph-21-01563-f004:**
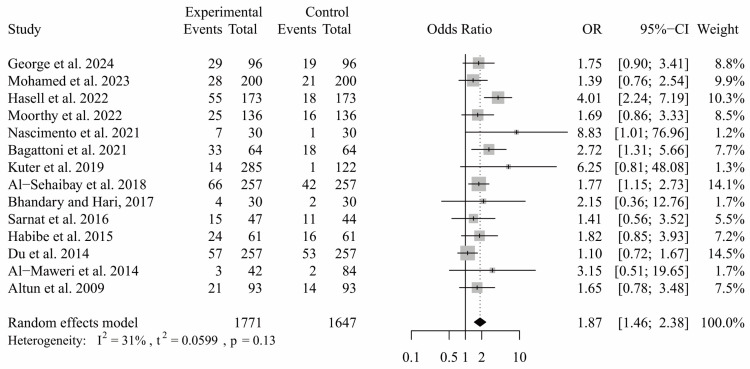
Meta-analysis comparing the incidence of TDI between patients with ASD and neurotypical patients following sensitivity analysis.

**Table 1 ijerph-21-01563-t001:** Characteristics of included studies (*n* = 22).

Author/Year	Study Design	Country	Number of Participants	Gender, *n*	Age (Mean or Range)	Setting of Study	Trauma Classification	TDI (%)	Key Findings
George et al., 2024 [25]	Cross-sectional	India	Autism: 96 Neurotypical: 96	Male: 116 Female: 76	3 to 5 years	University	WHO classification modified by Andrasen	Autism: 30.2% Neurotypical: 19.8%	ASD children showed increased TDI compared to neurotypical children.
Mohamed et al., 2023 [26]	Prospective Cohort	Egypt	Autism: 200 Neurotypical: 200	NR	6 to 12 years	University	WHO classification modified by Andrasen	Autism: 14% Neurotypical: 10.5%	Majority of the ASD children usually exhibited a higher occurrence of TDIs than neurotypical children.
Hasell et al., 2022 [27]	Retrospective Cohort	Canada	Autism: 173 Neurotypical: 173	Male: 250 Female: 96	6 to 14 years	University	NR	Autism: 31.8% Neurotypical: 10.4%	Children living with ASD showed significant TDI in comparison to neurotypical patients
Moorthy et al., 2022 [28]	Case-Control	India	Autism: 136 Neurotypical: 136	Male: 193 Female: 79	5 to 12 years	Special Schools	NR	Autism: 18.4% Neurotypical: 11.8%	Dental trauma was reported in more children from ASD as compared to the neurotypical patients; however the difference was not statistically significant
Nascimento et al., 2021 [29]	Cross-sectional	Brazil	Autism: 30 Neurotypical: 30	Male: 14 Female: 46	3 to 13 years	University	WHO classification modified by Andrasen	Autism: 23.3% Neurotypical: 3.3%	Children with ASD had a higher occurrence of dental trauma, when compared to neurotypical children
Santosh et al., 2021 [30]	Cross-sectional	India	Autism: 142 Neurotypical: -	Male: 114 Female: 28	3 to 17 years	Special Schools	Ellis Fracture	Autism: 13.8% Neurotypical: -	13.38% of ASD children exhibited TDI
Bagattoni et al., 2021 [31]	Cross-sectional	Italy	Autism: 64 Neurotypical: 64	Male: 79 Female: 49	Mean of 8.7 years	University	NR	Autism: 52% Neurotypical: 29%	Individuals with ADS are at greater risk of TDI and that epilepsy may be an additional risk factor
Basha et al., 2021 [32]	Cohort	Saudi Arabia	Autism: 74 Neurotypical: -	NR	Mean of 12 years	Special Schools	NR	Autism: 14.9% Neurotypical: -	Significant association between TDI prevalence and increased overjet, inadequate lip coverage
Baskar et al., 2020 [33]	Retrospective Cohort	India	Autism: 16 Neurotypical: -	Male: 10 Female: 6	6 to 18 years	University	Ellis Fracture	Autism: 37.5% Neurotypical: -	Children with ASD exhibit risk factors of fall to be the most common when compared to those neurotypical patients
Kuter et al., 2019 [15]	Cross-sectional	Turkey	Autism: 285 Neurotypical: 122	NR	5 to 16 years	NR	NR	Autism: 4.7% Neurotypical: 1%	The results showed a significant difference in TDI in patients with ASD than neurotypical patients
Al-Sehaibany et al., 2018 [34]	Cross-sectional	Saudi Arabia	Autism: 257 Neurotypical: 257	Male: 357 Female: 77	3 to 5 years	ASD centers kindergarten	WHO classification modified by Andrasen	Autism: 25.7% Neurotypical: 16.3%	The occurrence of TDIs was higher in Saudi preschool children with ASD than in neurotypical children
Bhandary and Hari 2017 [35]	Case-control	India	Autism: 30 Neurotypical: 30	Male: 28 Female: 32	6 to 12 years	Special Schools and Hospitals	NR	Autism: 13% Neurotypical: 6%	There was no significant difference in the occurrence of TDI between ASD and neurotypical children
Bagattoni et al., 2017 [36]	Cross-sectional	Italy	Autism: 46 Neurotypical: -	NR	Mean of 9.2 years	University	WHO classification modified by Andrasen	Autism: 30.4% Neurotypical: -	The prevalence of TDI in children and adolescents with ASD is high
Moralez-chavez et al., 2016 [37]	Cross-sectional	Venezuela	Autism: 94 Neurotypical: -	NR	4 to 16 years Means of 7.24 years	Dental Clinic	NR	Autism: 13.8% Neurotypical: -	TDI are more frequently found in patients with ASD
Andrade et al., 2016 [9]	Cross-sectional	Brazil	Autism: 114 Neurotypical: 114	Male: 114 Female: 114	3 to 15 years	University	NR	Autism: 24.6% Neurotypical: 41.2%	The prevalence of TDI was lower in ASD individuals compared to neurotypical patients
Sarnat et al., 2016 [38]	Case-control	Israel	Autism: 47 Neurotypical: 44	Male: 49 Female: 42	3.5 to 8 years	Kindergartens	NR	Autism: 32% Neurotypical: 25%	There was no significant difference in the occurrence of TDI between ASD and neurotypical children
Habibe et al., 2015 [14]	Case-control	Brazil	Autism: 61 Neurotypical: 61	Male: 98 Female: 24	4 to 7 years	University	WHO classification modified by Andrasen	Autism: 39% Neurotypical: 26%	There was no significant difference in the occurrence of TDI between ASD and neurotypical children
Du et al., 2014 [39]	Case-control	China	Autism: 257 Neurotypical: 257	Male: 217 Female: 297	2 to 6 years	Special Schools	WHO classification modified by Andrasen	Autism: 22.2% Neurotypical: 20.6%	There was no significant difference in the occurrence of TDI between ASD and neurotypical children
Al-Maweri et al., 2014 [40]	Case-control	Yemen	Autism: 42 Neurotypical: 84	Male: 99 Female: 27	6 to 15 years	Special Schools	WHO classification modified by Andrasen	Autism: 7.1% Neurotypical: 2.4%	There was no significant difference in the occurrence of TDI between ASD and neurotypical children
Orellana et al., 2012 [16]	Case-control	Spain	Autism: 30 Neurotypical: 30	Male: 50 Female: 10	20 to 41 years Mean of 27.7 years	Special Schools	NR	Autism: 20% Neurotypical 30%	There was no significant difference in the occurrence of TDI between ASD and neurotypical children
Ferreira et al., 2010 [41]	Retrospective Cohort	Brazil	Autism: 11 Neurotypical: -	NR	8.1 years	University	NR	Autism: 27.3% Neurotypical: -	There was no significant difference in the occurrence of TDI between ASD and neurotypical children
Altun, 2009 [42]	Cross-sectional	Turkey	Autism: 93 Neurotypical: 93	Male: 138 Female: 48	4 to 23 years	University	WHO classification modified by Andrasen	Autism: 23% Neurotypical: 15%	There was no significant difference in the occurrence of TDI between ASD and neurotypical children

ASD: Autism Spectrum Disorders; NR: Not Reported; TDI: Traumatic Dental Injuries; Without Neurotypical Group.

**Table 2 ijerph-21-01563-t002:** Quality assessment of included studies.

Study/Year	Was this Study Question or Objective Clearly Stated?	Was this Study Population Clearly and Fully Described, Including a Case Definition?	Were the Cases Consecutive?	Were the Subjects Comparable?	Was the Intervention Clearly Described?	Were the Outcome Measures Clearly Defined, Valid, Reliable, and Implemented Consistently Across All Study Participants?	Was the Length of Follow-Up Adequate?	Were the Statistical Methods Well-Described?	Were the Results Well-Described?	Total (*n*/9)
George et al., 2024 [25]	0	1	1	1	0	1	1	1	1	7/9
Mohamed et al., 2023 [26]	1	0	1	1	1	1	1	0	1	7/9
Hasell et al., 2022 [27]	1	1	1	1	0	1	0	1	1	7/9
Moorthy et al., 2022 [28]	1	1	1	1	1	1	0	1	1	8/9
Nascimento et al., 2021 [29]	1	1	1	1	1	1	1	1	1	9/9
Santosh et al., 2021 [30]	1	1	0	0	1	1	1	0	1	6/9
Bagattoni et al., 2021 [31]	1	1	0	1	0	1	1	1	1	7/9
Basha et al., 2021 [32]	1	1	0	0	1	1	1	0	1	6/9
Baskar et al., 2020 [33]	1	0	0	0	1	1	0	1	1	5/9
Kuter et al., 2019 [15]	0	1	0	1	0	1	1	1	0	5/9
Al-Sehaibany et al., 2018 [34]	1	1	1	1	1	1	1	1	1	9/9
Bhandary and Hari 2017 [35]	0	1	1	1	1	1	0	1	1	7/9
Bagattoni et al., 2017 [36]	1	1	0	0	1	1	1	1	1	7/9
Moralez-chavez et al., 2016 [37]	1	1	1	0	1	0	1	0	0	5/9
Andrade et al., 2016 [9]	1	1	1	1	0	1	0	1	1	7/9
Sarnat et al., 2016 [38]	1	0	1	1	0	1	1	0	0	5/9
Habibe et al., 2015 [14]	1	1	1	1	1	1	1	1	1	9/9
Du et al., 2014 [39]	1	1	0	1	1	1	1	1	1	8/9
Al-Maweri et al., 2014 [40]	1	0	1	1	1	1	0	1	0	6/9
Orellana et al., 2012 [16]	0	0	1	1	0	1	1	1	0	5/9
Ferreira et al. [41]	1	0	1	1	0	1	0	1	1	6/9
Altun, 2009 [42]	1	1	1	1	1	1	1	1	1	9/9

## Data Availability

No new data were created for this review.

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
