# Peer review of "Increased Risk of Dentoalveolar Trauma in Patients with Autism Spectrum Disorder: A Systematic Review with Meta-Analysis"

_ijerph, 2024, doi:10.3390/ijerph21121563_

Round 1

Reviewer 1 Report

Comments and Suggestions for Authors

Dear author,

Line 59--Many of these injuries stem from preventable everyday accidents [9]- Explain the statement, referring to which preventable conditions.

Introduction:  Line 76, 78----The comparison was performed with non-ASD persons; the author did not address or discuss neurotypical individuals.  However, in the aim and hypothesis, it is mentioned.Please clarify whether it is solely ASD or both ASD and neurotypical individuals.  Explain the conditions to which neurotypical individuals are considered.

Line 163….Ultimately, each outcome's  certainty was rated as high, moderate, low, or very low. –Mention the numerical rating or score division, which ranges from high to very low. 

Result  section: 4 from the Cochrane Library (line 173). But in Search Strategies, Cochrane library is not mentioned (lines 113,114).

In several instances, neurotypical controls are stated. Clarify the conditions for the control group. ---There's no consistency is maintained.

 Line 247-250. The initial certainty was rated low, as only observational studies were included, and it was 248 further downgraded due to indirectness. No upgrades were applied for large  effects, plausible confounding, or dose-response gradients. What is the paragraph intended to clarify?

225- The objective of this study was to evaluate the prevalence of TDI in patients with ASD and to compare it to that of non-ASD patients, thereby determining whether there were neurotypical or non-ASD controls.

It is referred to as neurotypical controls in the hypothesis and non-ASD in the discussion.  The terminology's meaning is confusing. 

Line 313-326 ----The author should address the prevalence of the condition by region, as Brazil and India had more published data, as previously mentioned in the results. The only data reported in the discussion pertains to Brazil; what about other countries?

It is necessary to include the prevalence and incidence of the age categories that were observed.

Comments on the Quality of English Language

The authors may contemplate a moderate degree of language correction.

Author Response

Thank you very much for the thorough evaluation and detailed revision of our manuscript. We appreciate the reviewer’s insightful comments. All modifications have been highlighted in red for clarity, while changes made to improve the English have been highlighted in bright blue. All responses to the reviewer’s questions are available in the attached PDF file (Detailed Response - Reviewer 1).

Reviewer 2 Report

Comments and Suggestions for Authors

This is an interesting review that addresses a relatively unexplored area. There are a few minor issues that should be addressed to enhance the article's clarity and rigor:

  • PROSPERO Registration:
    • The information at the PROSPERO platform needs to be updated. 
    •  
  • Abstract:
    • Some information is missing from the abstract to fully comply with the PRISMA checklist:
      • Specify inclusion and exclusion criteria 
      • Specify the methods used to present and synthesise results.
      •  
  • PECOS Description:
    • The PECOS (Population, Exposure, Comparison, Outcomes, Study design) framework used for this review should be explicitly described to clarify the study’s scope and improve reproducibility.
  • Data Extraction (Points 9 and 10 - TDI and Findings):
    • In the data extraction section, points 9 and 10 (TDI and Findings) need further explanation. Specifically, clarify whether the TDI prevalence was calculated independently by the authors or directly extracted from the included studies.
  • Results:
    • There are discrepancies in the gender data for Basha et al. (2021) and Bagattoni et al., as the numbers do not align with the participant totals in the right column. Please double-check and correct these figures.
    • In lines 228–229, it’s mentioned that results favor ASD participants. Please clarify how the results favor this group.
  • Citation:
    • In the statement, “According to a previous study, the global prevalence of TDI is approximately 10–15%,” the original study should be cited, Petti (2018).

Author Response

Thank you very much for the thorough evaluation and detailed revision of our manuscript. We appreciate the reviewer’s insightful comments. All modifications have been highlighted in red for clarity, while changes made to improve the English have been highlighted in bright blue. All responses to the reviewer’s questions are available in the attached PDF file (Detailed Response - Reviewer 2).

Round 2

Reviewer 1 Report

Comments and Suggestions for Authors

Dear Authors,

Regards for correcting as per the suggestions.